# The interplay at the replisome mitigates the impact of oxidative damage on the genetic integrity of hyperthermophilic *Archaea*

Tom Killelea[1], Adeline Palud[1], Farida Akcha[2], Mélanie Lemor[1], Stephane L'haridon[1], Anne Godfroy[1], Ghislaine Henneke[1]*

[1]Univ Brest, Ifremer, CNRS, Laboratoire de Microbiologie des Environnements Extrêmes, Plouzané, France; [2]Laboratoire d'Ecotoxicologie, Ifremer, Nantes, France

**Abstract** 8-oxodeoxyguanosine (8-oxodG), a major oxidised base modification, has been investigated to study its impact on DNA replication in hyperthermophilic *Archaea*. Here we show that 8-oxodG is formed in the genome of growing cells, with elevated levels following exposure to oxidative stress. Functional characterisation of cell-free extracts and the DNA polymerisation enzymes, PolB, PolD, and the p41/p46 complex, alone or in the presence of accessory factors (PCNA and RPA) indicates that translesion synthesis occurs under replicative conditions. One of the major polymerisation effects was stalling, but each of the individual proteins could insert and extend past 8-oxodG with differing efficiencies. The introduction of RPA and PCNA influenced PolB and PolD in similar ways, yet provided a cumulative enhancement to the polymerisation performance of p41/p46. Overall, 8-oxodG translesion synthesis was seen to be potentially mutagenic leading to errors that are reminiscent of dA:8-oxodG base pairing.
DOI: https://doi.org/10.7554/eLife.45320.001

*For correspondence: ghenneke@ifremer.fr

Competing interests: The authors declare that no competing interests exist.

## Introduction

Oxidative stress, arising as a result of a disturbance between the production of reactive oxygen and nitrogen species (ROS and RNS), and antioxidant defences, is associated with damage to a wide range of molecular species including lipids, proteins, and nucleic acids (*McCord, 2000*; *Imlay, 2003*; *Apel and Hirt, 2004*). 8-oxodeoxyguanosine (8-oxodG) is one of the major oxidised bases in DNA or the nucleotide pool (*Kamiya, 2003*). Base damage is caused by highly reactive free radicals (*Kasai and Nishimura, 1984*; *Burrows and Muller, 1998*; *Neeley and Essigmann, 2006*; *Cadet et al., 2008*) generated exogenously upon exposure to ionising radiation, environmental factors (*e.g.*, transition metals, chemicals, free radicals, etc), or endogenously from metabolic processes in different cellular compartments (*e.g.*, peroxisomes, mitochondria, chloroplasts, etc) (*Krumova and Cosa, 2016*).

To protect against 8-oxodG accumulation and potential mutagenesis, cells evolved enzymatic repair mechanisms that ensure both the removal of the oxidised deoxyribonucleoside (8-oxodG) from genomic DNA and the degradation of the oxidised DNA precursor (8-oxodGTP), thereby preventing its incorporation by DNA polymerases (DNA pols) (for review, see *David et al., 2007*; *Kamiya, 2010*). Evolutionarily conserved from *Bacteria* to eukaryotes, the repair of 8-oxodG in DNA utilises the base excision repair (BER) pathway, ensuring the removal of dC:8-oxodG and dA:8-oxodG mispairs respectively by OGG1/MutM (Fpg) and MUTYH/MutY BER glycosylases in eukaryotic/*E. coli* cells. While demonstrated only in eukaryotes, other defence mechanisms such as the mismatch repair (MMR), nucleotide excision repair (NER) and transcription coupled-NER (TC-NER) may

function as effective substitutes for 8-oxodG removal (*Tuo et al., 2002*; *Russo et al., 2004*; *Macpherson et al., 2005*).

Although most 8-oxodG damage is repaired by these preventive systems (for review see *van Loon et al., 2010*), 8-oxodG that escapes repair is likely to be encountered by DNA pols during either replicative or repair DNA synthesis. The extent to which 8-oxodG is bypassed depends on the identity of the prokaryotic and eukaryotic DNA pols. The nucleotides dAMP and dCMP are incorporated opposite template 8-oxodG to varying efficiencies, potentially causing dG→dT transversion mutations during subsequent rounds of DNA replication (*Hübscher and Maga, 2011*; *Berquist and Wilson, 2012*). The differences in selectivity for nucleotide insertion are dictated by the intrinsic features of DNA pols (active site steric constraints, specific interactions with the backbone of the template 8-oxodG, etc), the sequence context in the genome (*Zahn et al., 2011*) and the modulating role of accessory factors (*Maga et al., 2007*; *Maga et al., 2008*; *Locatelli et al., 2010*). The premutagenicity of 8-oxodG in DNA is mainly due to its Hoogsteen base pairing in the *syn* conformation with dA (*Chemical structure 1*) and the ability of DNA pols to extend the resulting mismatch (*Shibutani et al., 1991*). Mimicking the geometry of a correct base pair, the dA:8-oxodG *anti:syn* mispair thus escapes the proofreading 3′–5′ exonuclease activity in the replicative polymerase (*Brieba et al., 2004*; *Hsu et al., 2004*).

dG(anti):dC(anti)          dT(anti):dA(anti)

Hoogsteen base pair

8-oxodG(anti):dC(anti)          8-oxodG(syn):dA(anti)

**Chemical structure 1.** Base pairing of 8-oxoguanosine.
DOI: https://doi.org/10.7554/eLife.45320.002

While the oxidation of the deoxyguanosine and its impact on the genome stability of aerobic organisms has been extensively documented in *Bacteria* and eukaryotes, there are limited reports about its occurrence and effect on archaeal cells. *Archaea,* the third domain of life, are represented by aerobic and anaerobic microorganisms that all are equipped with ROS removal systems, indicating their appearance early in the evolution of life (*Wiedenheft et al., 2005*; *Halliwell, 2006*; *Ramsay et al., 2006*). Thriving in hostile habitats (such as hydrothermal vents, cold seeps, springs and salt lakes) under harsh environmental conditions (such as elevated temperature, high pressure, pH shifts, heavy metals, ionising radiations, etc) it is theorised that *Archaea* face large-scale DNA damage, thereby challenging replication accuracy. Examined in few aerobic euryarchaeal and crenarchaeal strains (two extreme halophiles *Haloferax Volcanii* and *Halobacterium salinarum* sp. NRC-1, and the thermoacidophile *Sulfolobus acidocaldarius*), the mutation frequencies were found to be comparable to those of mesophilic microbial genomes (*Grogan et al., 2001*; *Mackwan et al., 2007*; *Busch and DiRuggiero, 2010*). Low rates of genomic mutation, such as those observed, suggest that these *Archaea* evolved molecular mechanisms to ensure their genome integrity. Conversely, the thermoacidophile *Sulfolobus solfataricus* exhibits an elevated rate of spontaneous mutations (one order of magnitude higher) which is mediated by transposition of insertion elements (*Martusewitsch et al., 2000*).

The hyperthermophilic anaerobic *Euryarchaea*, *Pyrococcus abyssi*, has been isolated from hydrothermal vents characterised by elevated temperatures, pH-shifts, radiation and differing metal concentrations (*Erauso et al., 1993*). It exhibits extreme resistance to ionising radiation (*Jolivet et al., 2003*) and can withstand a moderate level of genomic abasic sites damage (25 abasic sites per

100000 bp) (*Palud et al., 2008*). Used as model system to understand the molecular basis of DNA replication (*Myllykallio et al., 2000*; *Matsunaga et al., 2003*), *P. abyssi* encodes two replicative DNA polymerases, a B-family and a D-family, which have both been functionally and structurally characterized alone or in the presence of replication factors (*Henneke et al., 2005*; *Rouillon et al., 2007*; *Castrec et al., 2009*; *Gouge et al., 2012*; *Henneke, 2012*; *Masuda et al., 2015*; *Sauguet et al., 2016*; *Lemor et al., 2018*; *Raia et al., 2019*). Both PolD and PolB contain exonuclease domains and display high nucleotide selectivity (*Palud et al., 2008*), with PolB described as one of the most accurate and processive enzymes (*Dietrich et al., 2002*). These features make them ideally suited for accurate DNA synthesis in DNA replication and repair. Completing the repertoire of DNA polymerisation enzymes is the DNA polymerase/primase complex (p41/p46). Devoid of any proofreading $3'-5'$ exonuclease activity, it has been identified as an RNA priming enzyme at the replication fork, and a potential DNA repair enzyme capable of synthesising short-patches of DNA (*Le Breton et al., 2007*; *Jozwiakowski et al., 2015*; *Lemor et al., 2018*).

Previous studies showing the strong resistance of *P. abyssi* to gamma irradiation (*Jolivet et al., 2003*) which exerts molecular oxidative stress in anoxic conditions makes this strain an ideal model to analyse the response of oxidative attacks from another oxidising agent, in this case, oxygen. In this study, we determine the steady-state level of 8-oxodG in the genome of normal growing cells and after exposure to oxygen. We further analyse the consequence that this damage has on the damage-bypass properties of cell-free extracts, and the individual DNA replication proteins, PolB, PolD and the p41/p46 complex alone, or in the presence of accessory factors (Proliferating Cell Nuclear Antigen and Replication Protein A for PCNA and RPA respectively). Finally, we measure the intrinsic $3'-5'$ exonuclease activity of PolD and PolB alone or with accessory proteins of 8-oxodG mispairs. The potential mutagenicity of 8-oxodG in DNA and more generally genomic maintenance in *Archaea* are thus discussed.

## Results

### Rate of 8-oxodG in the genome of *P. abyssi*

Before analysing the *in vitro* properties of the DNA pols in the presence of 8-oxodG, we investigated whether this DNA lesion is present in the genome of *P. abyssi*, and how the levels compare to a mesophilic bacterial control, *E. coli*. The steady-state level of 8-oxodG for both organisms was calculated during the exponential and stationary phases of growth (*Table 1*). In the exponential phase, 63.2 8-oxodG/$10^6$ dG was calculated for the *P. abyssi* genome, with the value moderately increasing to 115.1 8-oxodG/$10^6$ dG at the stationary phase. Comparatively, 8-oxodG was not detectable in the genome of *E. coli* in both phases of growth.

**Table 1.** Rate of endogenous genomic 8-oxodG/$10^6$ dG in *P. abyssi* and *E. coli* genomes at different growth phases.

Steady-state level of 8-oxodG per $10^6$ dG was calculated during the exponential and stationary growth phases. The number of 8-oxodG per $10^6$ dG represents the average of triplicate experiments from two biological samples with the standard deviation (±) shown. ND means No Detectable (ND is assigned to values below the HPLC-EC-UV detection limit of 0.01 pmol of 8-oxodG). Raw data are presented in *Table 1—Source data 1*.

| | Growth phase | 8-oxodG/$10^6$ dG |
|---|---|---|
| *E. coli* | Exponential | ND |
| | Stationary | ND |
| *P. abyssi* | Exponential | 63.2 ± 4.6 |
| | Stationary | 115.1 ± 5.8 |

DOI: https://doi.org/10.7554/eLife.45320.003

The following source data is available for Table 1:

Source data 1. Quantification of the steady-state level of 8-oxoguanosine in the genome of E.coli and P.abyssi.
DOI: https://doi.org/10.7554/eLife.45320.004

Colonising hydrothermal vents *P. abyssi* encounters environmental fluctuations and has to deal with numerous genotoxic events (*Huber et al., 1990*; *Summit and Baross, 1998*). Here, proliferating *P. abyssi* were aerated to induce oxidative stress. Oxygen sparging of exponentially growing culture for 5 min gave rise to increased levels of genomic 8-oxodG (*Figure 1*). The level (174.9 8-oxodG/$10^6$ dG at time T1) was approximately 3-fold higher than observed before air exposure (60.1 8-oxodG/$10^6$ dG at time T0). After 5 min (time T1), approximately 12% of the cells were viable as observed by a lower cell density ($7.5 \times 10^6$ cells/mL compared with $9.3 \times 10^7$ cells/mL at time T0). After 40 min (time T2) few oxygen-resistant cells were detectable ($4.3 \times 10^4$ cells/mL), which unfortunately did not allow the quantification of 8-oxodG because of insignificant amount of genomic DNA. After 140 min (time T3), the number of cells increased to $9.3 \times 10^4$ cells/mL. Concomitantly with active cell proliferation, a complete recovery of the basal level of 8-oxodG was obtained (65.1 8-oxodG/$10^6$ dG at time T3). After 320 min the basal level of 8-oxodG remained constant according to the mean $\pm$ standard deviation of 8-oxodG/$10^6$ dG measured at T0 and in *Table 1*. Taken together, these data provide the evidence for the first time that the hyperthermophile anaerobe *P. abyssi* possesses the necessary molecular mechanisms to overcome the presence of genomic 8-oxodG. Moreover, *P. abyssi* can withstand oxidative stress by counteracting and rapidly returning to the basal level of 8-oxodG in DNA.

## Replication bypass of template strand 8-oxodG by *P. abyssi* cell extracts

Previous studies undertaken in *E. coli* have determined that DNA damage bypass in the form of trans-lesion synthesis (TLS) can be observed using cellular extracts (*Wang et al., 1997*). In this work, DNA synthesis capable of bypassing DNA lesions was measured *in vitro* using *P. abyssi* cell-extracts (*Pab*CE) from exponentially growing cells (*Figure 2A*). Using a DNA substrate with a primer-

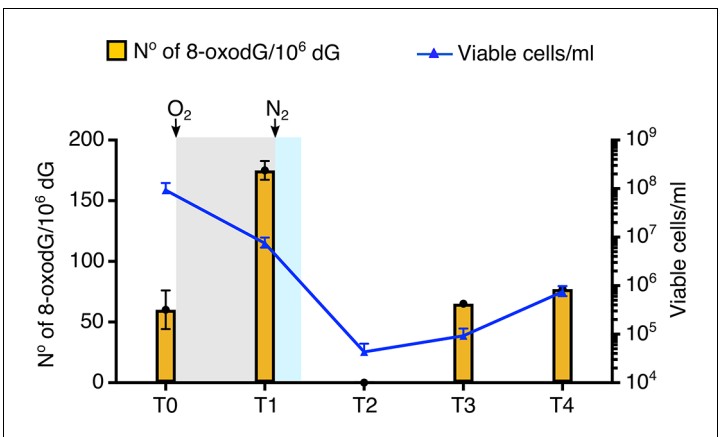

**Figure 1.** Effect of oxygenation on the viability and the rate of 8-oxodG/$10^6$ dG into the genome of *P. abyssi*. Oxidative stress is applied to *P. abyssi* growing cells in a batch mode culture during 5 min. Steady-state level of 8-oxodG per $10^6$ dG and viability are estimated at different times. T0, control before oxidative stress. T1, after the 5 min oxidative stress. T2, 40 min after the 5 min oxidative stress. T3, 140 min after the 5 min oxidative stress. T4, 320 min after the 5 min oxidative stress. Steady-state level of 8-oxodG is calculated from 10 µg of genomic DNA by HPLC-UV-EC as described in the methods. Errors bars indicated analytical duplicates. To enumerate viable cells, most-probable-number (MPN) assays were performed as previously published (*Blodgett, 2006*) (*Oblinger and Koburger, 1975*) Survival (cells/ml) is based on a three-tube MNP dilution assay. Upper and lower error bars are shown. Gray and blue shaded indicates the presence of dissolved oxygen in the medium culture. White background corresponds to strict anaerobia. Sparging with oxygen or nitrogen is shown with an arrow. Raw data for each graph are provided in *Figure 1—source data 1*.

DOI: https://doi.org/10.7554/eLife.45320.005

The following source data is available for figure 1:

**Source data 1.** Quantification of the effect of oxygenation on the viability of P. abyssi cells and the rate rate of 8-oxodG/106 dG in the genome.

DOI: https://doi.org/10.7554/eLife.45320.006

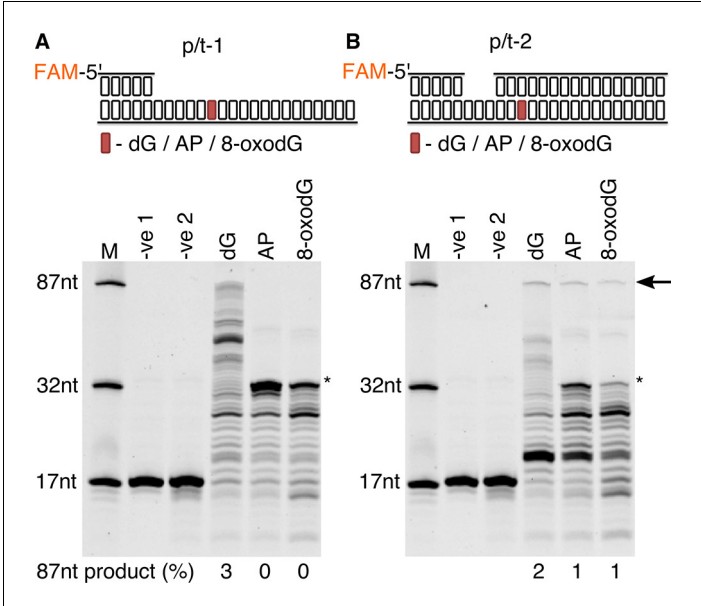

**Figure 2.** Primer extension and replication bypass of template strand 8-oxodG by *P. abyssi* cell extracts. Primer-template extension carried out for 60 min on p/t-1 containing either dG/AP/8-oxodG (**A**) or p/t-2 containing either dG/AP/8-oxodG (**B**). In both instances –ve one indicates sample lacking *P. abyssi* cell extract and –ve two indicates sample containing cell extracts but lacking MgCl$_2$ and dNTPs. An arrow is used to indicate the position of full length extension products (87nt in length), with * used to highlight the approximate location of the damaged base. Shown above each gel is a representative cartoon indicating the structure of primer-templates and the relative position (+33) of the dG/AP/8-oxodG within both DNA primer-template (highlighted in red).
DOI: https://doi.org/10.7554/eLife.45320.007
The following figure supplement is available for figure 2:

**Figure supplement 1.** Primer extension and endonuclease activity on 8-oxodG by *P. abyssi* cell extracts.
DOI: https://doi.org/10.7554/eLife.45320.008

---

template conformation (p/t-1), it was observed that *Pab*CE is capable of extending primers when presented with dG control in the template strand, with products ranging from 18 to 87nt in length (3% of primers being extended to full length (87nt)). However, when encountering both an abasic site (AP) and 8-oxodG in template strand DNA, a total arresting of DNA polymerisation one base upstream of the damage base (32nt) is observed. Rather than being an indication of TLS failure by the replication machinery, lack of primer extension was due to highly specific nuclease activity, resulting in cleavage of the phosphodiester bond of the damaged nucleotide before it was encountered by the polymerisation enzyme (*Figure 2—figure supplement 1*). Altering the DNA substrate to incorporate the 8-oxodG within a region of double stranded DNA (p/t-2) enabled low, yet visible levels of TLS by *Pab*CE, with 1% of primer DNA being fully extended in oxidative damage containing substrates, as opposed to 2% for the dG containing substrate (*Figure 2B*).

## Replication bypass of template strand 8-oxodG by replicative DNA proteins of *P. abyssi*

Next, we evaluated the ability of the three replicative enzymes from *P. abyssi* to bypass template strand 8-oxodG under running start conditions, as previously published (*Palud et al., 2008*). For each of the enzymes it was observed that encountering template strand 8-oxodG noticeably stalls primer extension compared to the dG control (*Figure 3A*), with each enzyme possessing a unique stall profile that is particularly apparent at the 10 min time point (*Figure 3B*). PolB arrests replication one nucleotide upstream of the 8-oxodG (32nt – 9%) or after having based paired a nucleotide opposite 8-oxodG (33nt – 9%). PolD stalls following base pairing opposite the lesion (33nt – 14%, *Figure 3B–C*) while also exhibiting downstream stalling after encountering 8-oxodG (7% at both 35 or 36nt position); of the two enzymes, the cumulative effect of 8-oxodG across all stall products

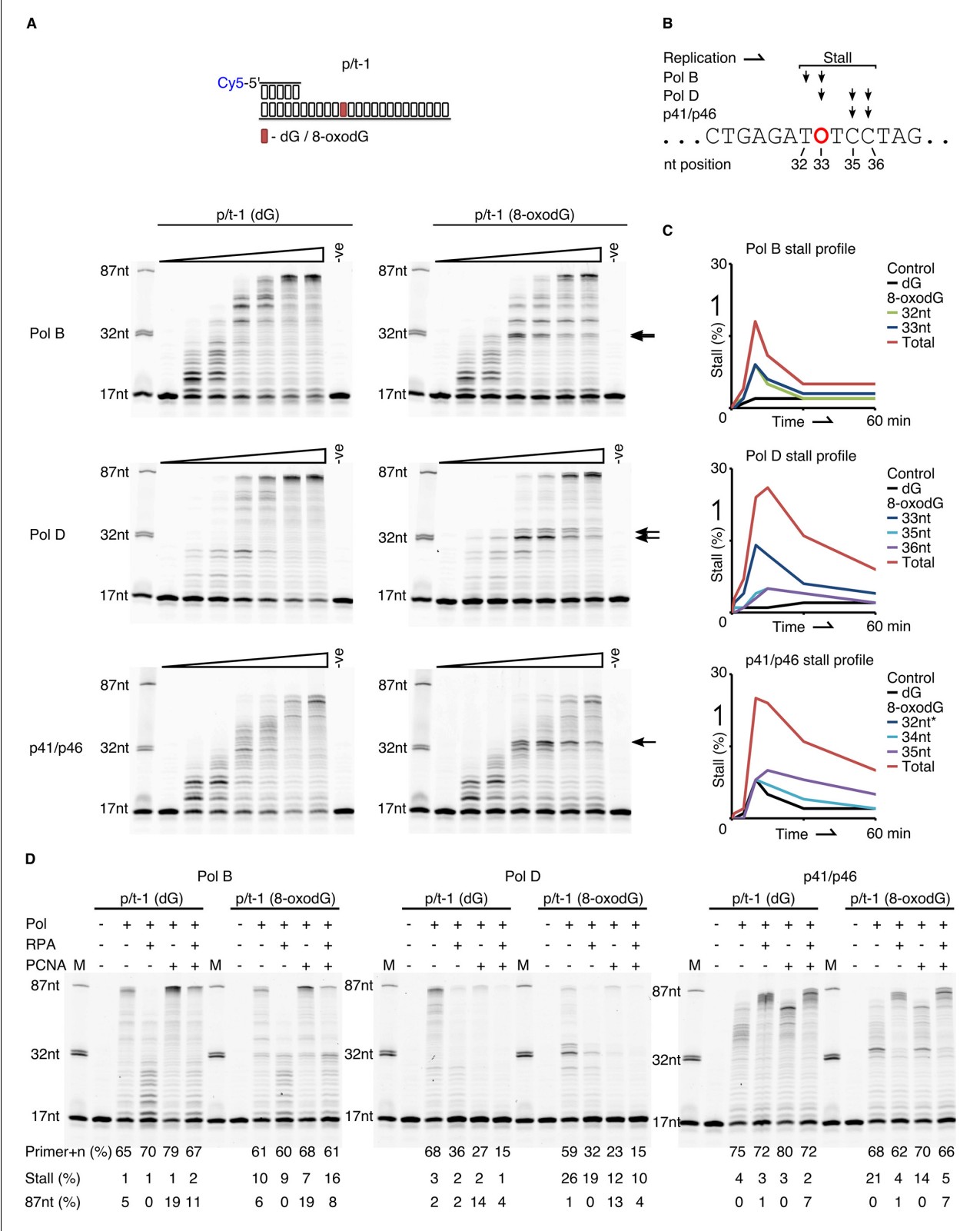

**Figure 3.** Primer extension and replication bypass of template strand 8-oxodG by *P. abyssi* replicative proteins. (**A**) Primer-template extension reactions performed on p/t-1 containing either dG or 8-oxodG by PolB, PolD and the p41/p46 complex. The triangle above each gel image indicates time course of primer extension (time points taken at 0, 0.5, 1, 5, 10, 30 and 60 min), the arrows to the right of each gel indicates the location of 8-oxodG induced stalling. –ve samples were incubated for 60 min lacking enzymes. (**B**) Diagrammatic representation of a section of p/t-1 template strand DNA indicating

*Figure 3 continued on next page*

*Figure 3 continued*

the positions at which the three replicative proteins stall replication at the 10 min time point, 8-oxodG is indicated by O. (**C**) Graphical representation of replication stalling profiles induced by 8-oxodG for each of the replicative proteins, shown for each individual stall event and as total 8-oxodG induced arrest at the 10 min time point (raw data for each graph are provided in *Figure 3—source data 1*).*-Data for p41/p46 8-oxodG 32nt stall are not visible due to mirroring that of the dG control. (**D**) Effects of RPA and PCNA on both primer extension and TLS activity of PolB, PolD and the p41/p46 complex during a 15 min reaction with p/t-1 containing either dG or 8-oxodG.

DOI: https://doi.org/10.7554/eLife.45320.009

The following source data and figure supplement are available for figure 3:

**Source data 1.** Qantification of gel bands arising from 8-oxodG induced stalling of the polymerisation enzymes from P. abyssi.
DOI: https://doi.org/10.7554/eLife.45320.013
**Figure supplement 1.** Primer extension under 'standing start' reaction conditions.
DOI: https://doi.org/10.7554/eLife.45320.010

appears to impact on PolD the most (*Figure 3C*). Despite the total impact of 8-oxodG induced blockage for p41/p46 at 10 min being of a similar level to that observed for PolD (*Figure 3C*), 8-oxodG itself has no observable impact on upstream primer extension or during incorporation opposite it. The enzyme partially arrests one nucleotide upstream of the 8-oxodG (32nt), but this is comparable to that observed for the dG control template, with more pronounced stalling seen at one (34nt) or two (35nt) nucleotides downstream of the damaged nucleotide (*Figure 3B–C*). Overall, these results demonstrate that *in vitro* both PolB and PolD exhibit prominent replication stalling upstream of template strand 8-oxodG, and while incorporating a nucleotide opposite the lesion, before eventually extending a primer beyond the lesion. In contrast to this, p41/p46 only stalls primer extension after bypassing the lesion.

We also examined the three DNA polymerisation enzymes under standing start conditions, with dG or 8-oxodG located in the +1 position from the primer-template junction (*Figure 3—figure supplement 1A*). The observed data correlate with that produced during running start bypass experiments in *Figure 3*; the p41/p46 complex, being least affected by 8-oxodG, with the percentage of primer+n decreasing by only 20% after 0.5 min when 8-oxodG is present, compared to dG. Primer extension by the two conventional replicative DNA pols is again affected by the presence of template strand 8-oxodG, with the levels of primer+n at 0.5 min decreasing by 41% and 70% for PolB and PolD respectively, in the presence of 8-oxodG compared to the dG control. Again, PolD and p41/p46 display stalling products one or two nucleotides downstream of the 8-oxodG lesion.

## Role of replication fork accessory proteins in replication bypass of template strand 8-oxodG

DNA polymerases are key components of the multi-protein replisome complex, working in conjunction with an array of partner proteins to engage in highly accurate DNA synthesis. As such, the functional interaction that occurs between the three replicative DNA enzymes (PolB, PolD and p41/p46) and two major replication fork proteins, RPA and PCNA was investigated (*Figure 3D*). Previous studies in *Euryarchaea* have observed a physical interaction between RPA and PolB (*Komori and Ishino, 2001*), PolD (*Komori and Ishino, 2001*; *Pluchon et al., 2013*), and the p41/p46 complex (*Komori and Ishino, 2001*; *Pluchon et al., 2013*). For PCNA both physical and functional interactions have previously been reported for PolB and PolD (*Rouillon et al., 2007*; *Castrec et al., 2009*). However, no interaction has been observed with the p41/p46 complex.

Here we report that in the presence of RPA primer extension by both PolB and PolD is initially inhibited, leading to an almost universal loss of full-length primer extension and shorter extension products over 15 min, with the same effect observed in the presence of 8-oxodG (*Figure 3D*). Under standing start conditions, with 8-oxodG placed at the +1 position from the primer-template junction, the influence of RPA is more pronounced causing primer utilisation to decrease by over 50% for both PolB and PolD (*Figure 3—figure supplement 1B*). Although RPA doesn't enhance primer utilisation by p41/p46, it does result in longer elongation products, partially mitigating the inhibition that the regulatory p46 subunit exhibits on extension activity (*Le Breton et al., 2007*). The presence of RPA largely alleviates any negative impact that 8-oxodG plays on stalling p41/p46 extension, with the previously observed downstream stalling removed in both running start and standing start extension reactions (*Figure 3—figure supplement 1B*).

PCNA also impacts on the primer extension activity all three polymerisation enzymes. Under running start conditions, PolD exhibits a reduction in primer+n formation in the presence of PCNA independently of 8-oxodG in the template strand (27% primer+n formation with PCNA compared to 68% without, for p/t-1 (dG) extension). However, this is offset by an increase in longer extension products (*Figure 3D*). On the other hand, primer utilisation by PolD does not seem impaired by PCNA in standing start experiments (*Figure 3—figure supplement 1B*). These results are in agreement with our previous data, showing that dNTP incorporation is less efficient on templates annealed to shorter DNA primers and that PCNA stimulates full-length DNA synthesis rather than primer utilisation by PolD (*Henneke et al., 2005*). PolB and p41/p46 exhibit increased primer elongation when reactions are supplemented with PCNA, corresponding to the accumulation of 87nt (PolB) or ~82 nt (p41/p46) products in length (*Figure 3D*). In the context of 8-oxodG bypass, all three polymerisation enzymes benefit from the addition of PCNA, with the % of stalled extension product being reduced, most likely due to the enhanced stability and processivity that PCNA provides (*Figure 3D*). These functional interactions are similarly observed under standing start experiments (*Figure 3—figure supplement 1B*).

The effect that both accessory proteins have on primer extension is noticeable (*Figure 3—figure supplement 1B*). For PolB the combined presence of both PCNA and RPA, while reducing the processivity enhancements that PCNA alone brings to PolB function, actually exacerbate PolB stalling in the presence of 8-oxodG (*Figure 3D*). For PolD the overall impact of both RPA and PCNA together is negative. This is most apparent in PolD when polymerisation is reduced to unobservable levels at 15 min. For both replicative DNA pols it is possible that additional replisome components are required to successfully mediate interactions with the RPA. Again, in contrast to the results observed with the two replicative DNA Pols, PCNA and RPA enhance the primer extension activity of the p41/p46 complex and remove any 8-oxodG induced downstream stalling (*Figure 3—figure supplement 1B*).

## Nucleotide insertion and extension at 8-oxodG template site by DNA polymerisation enzymes of *P. abyssi*

As 8-oxodG can be bypassed by each of the three enzymes, its mutagenic base-pairing potential was evaluated by single nucleotide incorporation experiments (*Figure 4*). In this

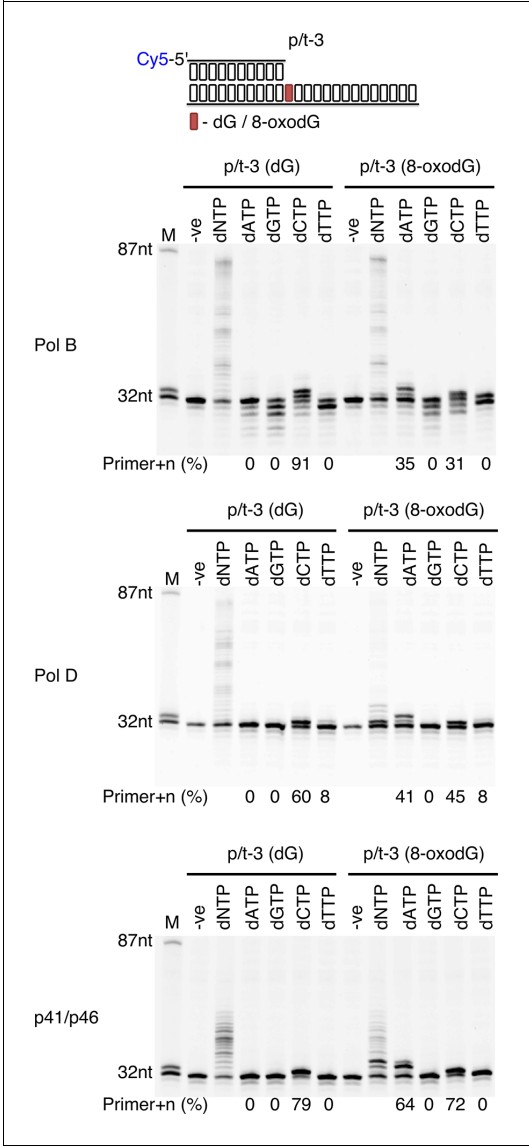

**Figure 4.** Single nucleotide incorporation opposite 8-oxodG by DNA polymerisation enzymes. Single nucleotide incorporation reactions were performed for each of the three replicative enzymes to determine the accuracy of incorporation when incorporating a single nucleotide opposite either dG or 8-oxodG when located at the +1 position from the primer-template junctions (p/t-3). All reactions were left for 5 min, with the –ve control lacking replicative enzyme.
DOI: https://doi.org/10.7554/eLife.45320.011

The following figure supplement is available for figure 4:

**Figure supplement 1.** Extension of primers pre-base paired opposite 8-oxodG.
DOI: https://doi.org/10.7554/eLife.45320.012

context, it is observed that all three DNA polymerisation enzymes provide correct incorporation opposite the dG control. Conversely, when incorporating opposite 8-oxodG all three enzymes are potentially mutagenic with dAMP and dCMP preferentially inserted. PolB suffers from a marked decrease in incorporation levels opposite 8-oxodG, with 35% and 31% primer+n for dAMP and dCMP respectively, compared to 91% dCMP incorporation opposite dG. PolD also shows a decreased incorporation when encountering 8-oxodG, 41% and 45% primer+n for dAMP and dCMP respectively, compared to 60% dCMP incorporation opposite dG. However, for the p41/p46 complex the levels of incorporation are similar regardless of the presence of oxidative damage, with 64% and 72% primer+n for dAMP and dCMP respectively, compared to 79% dCMP incorporation opposite dG.

To determine if the three polymerisation enzymes are efficient extenders of dA:8-oxodG or dC:8-oxodG paired termini, primer extension compared to correctly base paired dC:dG and the mismatched dT:dG. When engaging in primer-template extension with base paired dA:8-oxodG or dC:8-oxodG there is a noticeable difference between the enzymes containing the intrinsic 3′−5′ exonuclease activity and the p41/p46 complex (*Figure 4—figure supplement 1*). For PolB primer+n formation occurs regardless of the presence of 8-oxodG, with the lowest activity observed in the presence of dC:8-oxodG (52% primer+n) compared to dA:8-oxodG (73% primer+n) and the dC:dG control (87% primer+n) at the 0.5 min time point. PolD appears less tolerant of 8-oxodG base pairs, with primer+n formation after 0.5 min of 27% and 12% with dA and dC paired opposite 8-oxodG, compared to the dC:dG control (49% primer+n). The p41/p46 complex proves more tolerant of the presence of 8-oxodG paired termini, with high levels of primer+n formation at 0.5 min of 69% (dA:8-oxodG) and 65% (dC:8-oxodG), compared to 83% primer+n for the dC:dG control. Unlike PolB and polD, the p41/p46 complex was unable to extend a dT:dG mismatch.

## Effects of 8-oxodG-containing primer-template and oxidative base pairs on 3′−5′ exonuclease activity

To further elucidate the differing mechanisms by which PolB and PolD stall primer extension when encountering template strand 8-oxodG, the ability of the damaged nucleotide to stimulate exonuclease (exo) activity was evaluated (*Figure 5*). Regardless of the base contained in the +1 nt position from the primer-template junction there is a marked difference in the exonuclease activity of the two DNA pols (*Figure 5A*). PolB exo activity is highly stimulated, with a partial decrease observed when 8-oxodG is present in the +1 position from the primer-template junction (84% primer-n compared to 96% for dG after 5 min). However, for PolD there is only minor exonuclease activity, with no significant difference observed in the presence of a damaged nucleotide in the +1 position (7% primer-n for 8-oxodG compared to 5% for dG after 5 min).

To examine the effect of exonuclease activity on base-paired 8-oxodG, the primer length was extended by one nucleotide with either dA or dC base paired with 8-oxodG (*Figure 5B*). Encountering the mutagenic Hoogsteen base paired dA:8-oxodG causes stimulation of the PolB exonuclease rate, 72% primer-n degraded compared with 55% and 43% for non-mutagenic Watson-Crick base paired dC:8-oxodG and dC:dG after 5 min, respectively. Accordingly, the primer containing the single dT:dG mismatch confers the highest enhancement in the rate of 3′–5′ exonucleolysis (87% primer-n after 5 min). In contrast, low stimulation of the PolD exonuclease activity is observed in the presence of the Hoogsteen base paired dA:8-oxodG, 27% primer-n after 5 min compared to 31% primer-n for dC:8-oxodG and 26% dC:dG. On the other hand, the dT:dG mismatch significantly stimulated the rate of exo degradation by PolD (58% primer-n after 5 min). Taken together, the data indicate that the presence of Hoogsteen base pairing enhances the PolB exonucleolysis rate, while having a minor effect on PolD.

Not often considered, here we sought to examine the putative functional roles of PCNA and RPA on the exonuclease activity. It appears that the presence of the accessory proteins impacts on exonucleolysis by PolB and PolD (*Figure 5C*). Mirroring the alterations seen in extension reactions, RPA acts as an inhibitor of the exonuclease activity for both PolB and PolD, independently of the nature of the 3′-end primer termini (dA:8-oxodG, dC:8-oxodG, dC:dG and dT:dG). Indeed, a reduction in total primer utilisation results in longer degradation product. On the other hand, PCNA stimulates exonuclease activity of both PolB and PolD, with primer degradation accentuated, particularly for the dT:dG mismatch.

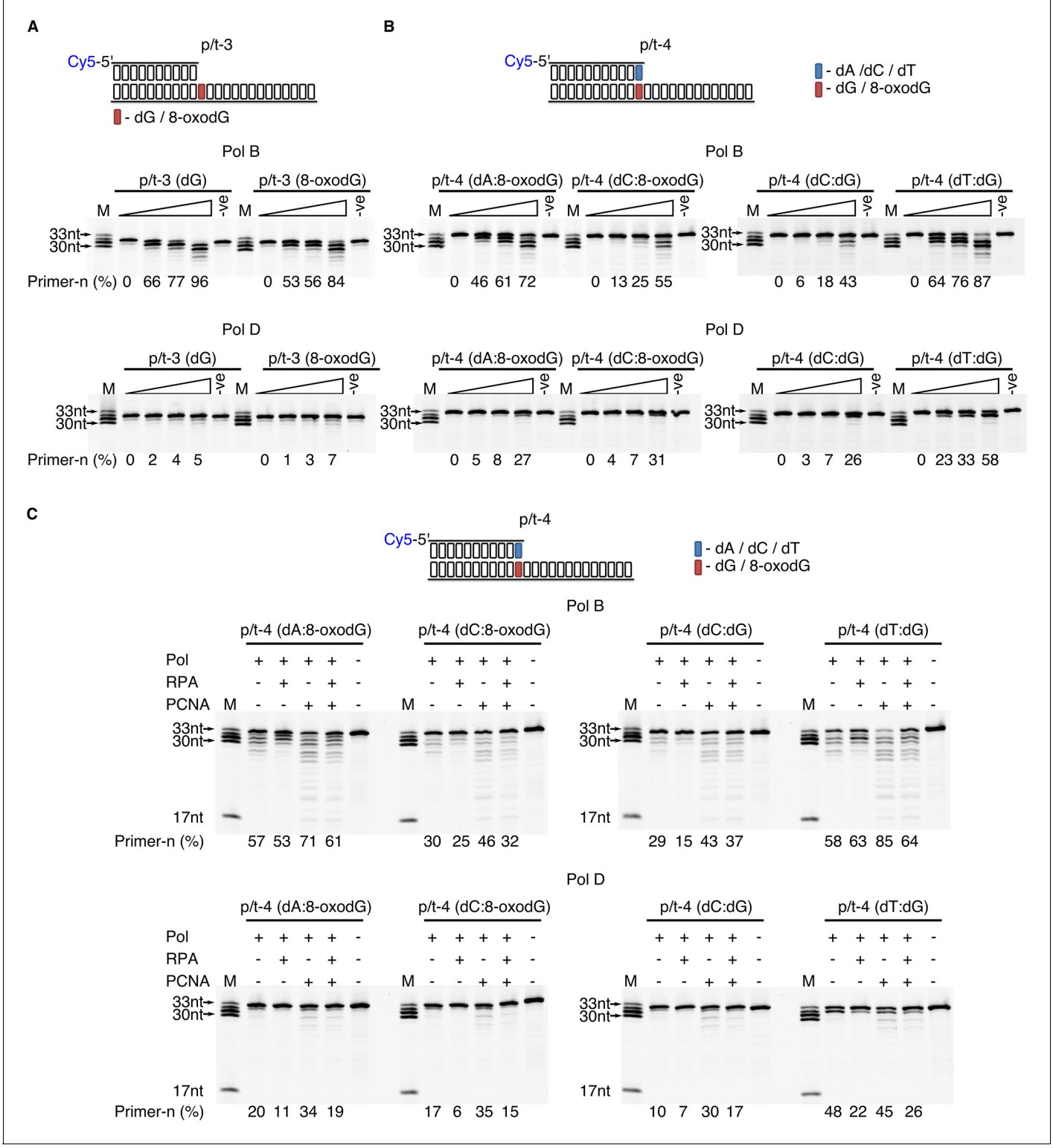

**Figure 5.** Effect of template strand 8-oxodG and 8-oxodG base paired termini on 3'—5' exonuclease activity. Exonuclease activity of PolB and PolD in the presence of dG and 8-oxoxdG when located in p/t-3,+1 nucleotides from the primer-template junction) (A) or when base paired with a complementary or non-complementary nucleotide in p/t-4, 0 nucleotide from the primer-template junction (B). In all instances, the triangle above the gel images denotes an exonuclease time-course with time points taken at 0, 0.5, 1 and 5 min. For all gels the –ve samples were incubated for 5 min lacking replicative enzyme. (C) Influence of RPA and PCNA on the exonuclease activity of PolB and PolD when encountering dG and 8-oxodG base paired with a complementary or non-complementary nucleotide (p/t-4) during a 5 min reaction.
DOI: https://doi.org/10.7554/eLife.45320.014

## Discussion

DNA replication is an inherently accurate process that relies on the successful interplay between replication and repair to maintain genome integrity in the face of numerous lesions, adducts and DNA roadblocks. In this study we have identified that growth conditions which mimic the natural habitat of deep sea hyperthermophilic anaerobic *Euryarchaeota* (*Erauso et al., 1993*; *Godfroy et al., 2006*) produce measurable levels of 8-oxodG in *P. abyssi* genomic DNA during both the exponential and stationary growth phases (*Table 1*), in line with previously published observations of alternative damage indicators (*Palud et al., 2008*). These results are also consistent with a recent study showing that the basal level of 8-oxodG in the genome of a closely related *Pyrococcus* species, *Thermococcus gammatolerans*, exceeds those usually measured in eukaryotic cells or *Bacteria* (*Barbier et al., 2016*). The level of 8-oxodG in the genome of *P. abyssi* is higher than that of *T. gammatorelans*, likely confirming that increased temperature of growth favours DNA oxidation (95°C compared with 85°C, respectively). Despite being highly sensitive to oxygen, a small fraction of *P. abyssi* cells survive stress and recover the steady-state level of 8-oxodG in 2.5 hr, consistent with survival rates observed following gamma-ray irradiation (*Jolivet et al., 2003*). The presence of basal levels of 8-oxodG in 'normal' growth conditions of hyperthermophilic anaerobic *Euryarchaeota* corroborates the existence of active DNA protection and repair mechanisms in proliferating cells of hyperthermophilic *Archaea* (*Barbier et al., 2016*).

In order to document any potential mechanism triggered in response to DNA oxidation in archaeal cells, primer extension past DNA lesions was measured *in vitro* using *Pab*CE from proliferating *P. abyssi*. In this instance TLS was only observed with the lesion present in a dsDNA conformation. Moreover, the lack of polymerisation past template strand 8-oxodG when present in a single stranded region of DNA is the result of specific nuclease activity which might be attributed to the 8-oxoguanine DNA glycosylase encoding gene in *P. abyssi*. This suggests that DNA repair mechanisms, such as BER, might be preferentially solicited over TLS, thus minimising the potential mutagenicity of 8-oxodG. It should be noted that BER has been assumed as a fundamental process in the removal of 8-oxodG in the genome of *Thermoccus gammatolerans* (*Barbier et al., 2016*). Thus, a preference for non-TLS repair is consistent with adduct bypass observations in model organisms, with TLS representing 8% of bypass events of replication hindering adducts in *Saccharomyces cerevisiae* (*Baynton et al., 1998*), and 1% (non-SOS induced) or 13% (SOS induced) in *E. coli* (*Koffel-Schwartz et al., 1996*).

Further credence for notion that TLS plays a limited role in DNA replication and repair can be derived from the mutagenic potential of 8-oxodG. Although both polymerases exhibit reduced nucleotide incorporation opposite 8-oxodG, PolB and PolD insert both dAMP and dCMP opposite the lesion (*Figure 4*). Similarly, p41/p46 inserts dAMP and dCMP opposite 8-oxodG but with the same efficiency as a canonical base control. Moreover, each of the three polymerisation enzymes extends dA:8-oxodG to a greater extent than dC:8-oxodG paired termini. This preferential extension may be due to the inability of the Pol active site to detect the mispair in the Pol active site since dA:8-oxodG Hoogsteen base pair mimics the geometry of a correct base pair (*Chemical structure 1*). The dC:8-oxodG paired termini appears strongly inhibitory to extension by PolD likely due to steric constraints incompatible with the size and flexibility of the polymerase active site cavity. Recent structural determination of PolD indicates that the Pol active site is not comparable to any known DNA polymerases, but rather shares similarities with RNA polymerases, despite DNA synthesising capabilities (*Sauguet et al., 2016*). One may expect that the Pol active site adopts specific unprecedented features when interacting with the primer-template and the incoming dNTP.

Possible scenario for lowering the mutagenic potential of the resulting 8-oxodG mismatches may involve MMR or $3'-5'$ exonuclease activity. Although recent work identified potential MMR proteins in Pyrococcus species (*Ishino et al., 2016*), here we focus on correction process by $3'-5'$ exonuclease activity. Relevant here is the finding that 8-oxodG mispairs are corrected by the exonuclease activities of PolD and PolB. Contrary to studies showing that *E. coli* Pol I (*Shibutani et al., 1991*), T7 Pol (*Brieba et al., 2004*), RB69 Pol (*Zhong et al., 2008*) and Pol δ (*McCulloch et al., 2009*) do not efficiently correct dA:8-oxodG mispair, significant exonuclease activity is observed with dA:8-oxodG and dC: 8-oxodG mismatches by PolD and PolB. It is interesting to notice that PolD exonuclease primer degradation is less active than PolB in the conditions tested. The PolD Exo active site exhibits structural dynamics clearly different to those of PolB in the presence of mismatches. This hypothesis

is corroborated by the unusual structural exonuclease domain of PolD which looks more similar to the Mre11 nuclease protein than the three characteristic sequence motifs termed Exo I, Exo II and Exo III (*Blanco et al., 1991*). *P. abyssi* HiFi (High Fidelity) enzymes may have evolved acute removal of mutagenic and non mutagenic primer termini due to the lack of appropriate extrinsic correction mechanisms. Not often considered, PCNA appears essential in exonuclease stimulation of mispairs by PolD and PolB, while RPA does not severely affect the activity.

Each of the three known polymerisation enzymes from *P. abyssi* possesses a unique stalling profile when encountering template strand 8-oxodG *in vitro*. Both PolB and PolD are observed to be sensitive to the presence of oxidative damage, stalling replication before eventually bypassing the lesion, whilst p41/p46 bypasses with high efficiency before stalling immediately downstream, consistent with previously published results (*Jozwiakowski et al., 2015*). However, the functional interactions observed between the polymerisation enzymes, RPA and PCNA enables further elucidation of the mechanism of TLS and the interplay that occurs in the archaeal replisome.

RPA enhances primer extension by the *P. abyssi* p41/p46 complex, functioning in a similar manner to eukaryotic RPA, which stimulates Pol α activity and processivity (*Braun et al., 1997*; *Maga et al., 2008*). The importance of RPA, particularly in the context of TLS, is highlighted by the manner in which it is able to alleviate downstream stalling of p41/p46. The presence of 8-oxodG in ssDNA and dsDNA results in alterations in the phospho-deoxyribose structure, causing variations to vertical base staking in the immediate vicinity of 8-oxodG (*Malins et al., 2000*). Such an effect is still present in dsDNA, but the presence of the complementary strand introduces a stiffening and stabilising effect, reducing the impact of 8-oxodG on DNA structure (*Crenshaw et al., 2011*). Similarly, RPA introduces rigidity to ssDNA (*Chen et al., 2015*) which in this instances negates the disruptive influence of 8-oxodG on ssDNA, reducing downstream stalling of the p41/p46 complex.

It is through observing the interactions between key components of the replisome and p41/p46 that we can clarify the nature of this enzyme as both a primase (*Figure 6A*) and a TLS polymerase that can bypass oxidative damage (*Figure 6B*). The p41/p46 complex is classified as an Archaeo-Eukaryotic Primase (AEP)(*Guilliam et al., 2015a*). The identification of a second human AEP (Prim-Pol) (*García-Gómez et al., 2013*), has instigated a revival in AEP enzyme research, that has at times resulted in a blurring of the lines amongst AEPs, particularly when discussing the *Archaeal* AEPs (*Guilliam et al., 2015b*; *Guilliam et al., 2017*). *In vitro*, human PrimPol can facilitate replication fork progression through TLS and downstream priming (*García-Gómez et al., 2013*). However, more recent studies evaluating the interplay with replisome partners highlight that RPA is able to both recruit (*Guilliam et al., 2017*) and regulate PrimPol activity, inhibiting replication on short primer-templates or when RPA is present in saturating conditions (*Guilliam et al., 2015b*; *Guilliam et al., 2017*; *Martínez-Jiménez et al., 2017*). This is in stark contrast to the data generated here, in which RPA enhances processivity of p41/p46, in a manner consistent with that observed for both Pol α (*Kenny et al., 1989*; *Braun et al., 1997*), the Pol α-Primase complex (*Dornreiter et al., 1992*) and Pol λ (*Krasikova et al., 2008*). In a further, more abrupt contrast, to date, PrimPol has no observable physical or functional interaction with PCNA (*Guilliam et al., 2015b*).

It is the interaction with PCNA that cements the duality of p41/p46 as a primase and TLS polymerase, enhancing the notion of divergent roles for the differing AEP clades. While still not fully understood, evidence from the eukaryotes suggests that replication factor C (RFC), the clamp loader, is instrumental in polymerase switching, displacing the low fidelity Pol α complex following primer synthesis and initial elongation and recruiting the high fidelity Pol δ during PCNA loading (*Waga and Stillman, 1994*; *Maga et al., 2000*; *Mossi et al., 2000*). While the exact role of RFC in *Archaea* is still contentious, there is evidence to suggest that in the *Crenarchaea S. solfataricus* it functions in a similar manner to its eukaryotic counterpart (*Wu et al., 2007*). If p41/p46 does not interact with PCNA during replication initiation, then when does the interaction occur? Y-family DNA polymerases Pol η, Pol ι and Pol κ, all TLS polymerases, have well-established physical and functional interactions with PCNA (*Bienko et al., 2005*; *Masuda et al., 2015*). Moreover, PCNA has been observed to stimulate polymerase activity and recruit the human TLS polymerase Pol ι to the replisome (*Haracska et al., 2001*), similar to the role observed in this paper.

In conclusion we have identified that as well as being a capable TLS polymerase that is stimulated by the stabilising presence of RPA, p41/p46 functionally interacts with the replisome 'scaffold' protein PCNA. Coupled with data highlighting that PolB and PolD stall upon encountering 8-oxodG in the presence of PCNA, this allows elucidation of the processes that occur when the replisome

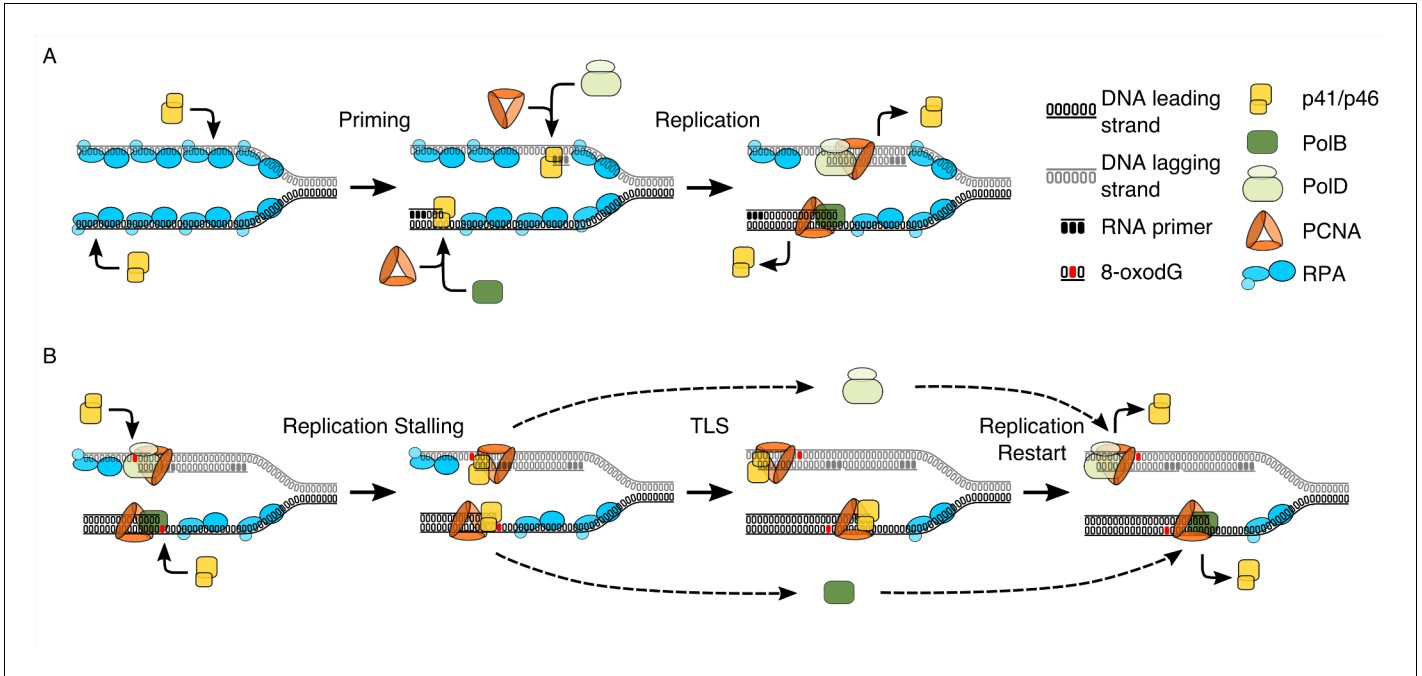

**Figure 6.** Model proposing the dual functionality of p41/p46 complex based on protein function and known interactions. (**A**) Priming: p41/46 is recruited at the replication fork and engages in the synthesis of a short RNA primer in a manganese catalysed reaction on both the leading and lagging strand. On the leading strand switching to a catalytic magnesium ion occurs, resulting in incorporation of dNTPs and extension of the RNA primer by p41/p46 until the loading of PCNA and the recruitment of PolB displace it to initiate DNA synthesis. Synthesis of an RNA primer by p41/p46 leads to direct loading of PCNA and recruitment of PolD to initiate DNA synthesis on the lagging strand (*Henneke et al., 2005*; *Le Breton et al., 2007*). (**B**) TLS: DNA replication stalls upon the HiFi Pol active site encountering oxidative damage. Unable to bypass the damaged nucleobase, polymerase switching occurs, with p41/p46 recruited to the primer-template junction and retained by PCNA. RPA bound to the single stranded region of DNA stabilizes the phosphodiester backbone downstream of the damage, allowing p41/p46 to function as a TLS polymerase, replicating past the damaged nucleobase until it is eventually replaced with a processive HiFi DNA polymerase.

DOI: https://doi.org/10.7554/eLife.45320.015

encounters a lesion such as 8-oxodG (*Figure 6B*). Replicative polymerases stall when encountering a lesion, possibly reversing replication with their exonuclease activity, this allows displacement of the polymerase and recruitment of the p41/p46 complex by the replisome 'scaffold' protein PCNA. Stabilised by RPA TLS occurs before p41/p46 is in turn displaced by one of the replicative DNA polymerases.

## Materials and methods

### Strains and cell culture techniques for 8-oxodG detection

The *E. coli* CIP 54.8 strain (Biological Resource Center of Institut Pasteur, https://research.pasteur.fr/fr/team/biological-resources-center/) was cultivated in 1 L of LB at 37°C with shaking. Cells have been collected in exponential and stationary growth phases by monitoring cell growth using cell counts. Cell suspensions were centrifuged for 30 min at 6000 g at 4°C and cell pellets were stored at –20°C.

*P. abyssi* GE5 (Brittany Culture Collection, http://www.ifremer.fr/souchotheque) culture experiments were performed under anaerobic conditions using a nitrogen-sparged gas-lift bioreactor as described (*Godfroy et al., 2006*). Exponentially growing cultures were obtained at 95°C on a complex SME medium at pH 6.5 at a dilution rate of 0.2 h$^{-1}$ (*Godfroy et al., 2000*). Stationary batch cultures were obtained by drawing off the culture from the bioreactor in order to obtain lower initial cell densities. Growth under batch conditions was followed by regular cell counting (every 15 min, three counts per samples) in order to collect stationary phase cells (*Postec et al., 2005*). Both exponentially and stationary phase cells were harvested on ice in bottles flushed with nitrogen before

use. Cold cell suspensions were transferred in oxygen impermeable leak free centrifugal devices in anaerobic glove box. Centrifugation was carried out for 30 min at 8000 g at 4°C. After elimination of the supernatant in the anerobic glove box, cell pellets were stored at –20°C. Thus, strict anaerobia (collection, centrifugation and storage) was maintained for *P. abyssi* before DNA extraction.

Cell densities were determined by direct cell counting using a Thoma cell (0.02 mm depth) under a phase contrast Olympus model BH-2 microscope (*Postec et al., 2005*).

## Genomic DNA extraction and digestion

Genomic DNA from *P. abyssi* and *E. coli* were isolated using the chaotropic NaI method (*Helbock et al., 1998*; *Akcha et al., 2000*; *Wessel et al., 2007*). Cell pellets were suspended in 2 mL lysis buffer (100 mM Tris-HCl pH 8.0, 1.4 M NaCl, 1% (w/v) SDS, 1% (v/v) β-mercaptoethanol (pure liquid; 14.3 M), 0.125 mM deferoxamine mesylate and disrupted with a Potter-Elvehjem homogenizer. Following centrifugation (5000 g for 5 min at 4°C), cell pellets were recovered and suspended in 800 μL of lysis buffer containing 2.5% (w/v) CTAB (Hexadecyltrimethylammonium bromide). The samples were then incubated at 65°C for 60 min. Following addition of 800 μL of chloroform:isoamyl alcohol (24:1), the tubes were gently mixed and centrifuged at 12000 g for 20 min at 4°C. The upper aqueous phases (~800 μL) were recovered and RNA digestion was performed by incubation with 20 μg of RNase A for 30 min at 37°C. Following addition of 1.2 mL of sodium iodide solution (20 mM EDTA-Na$_2$, 7.6 M NaI, 40 mM Tris–HCl, 0.3 mM deferoxamine mesylate, pH 8.0), the tubes were centrifuged at 10000 g for 20 min at 4°C. The pellets were then recovered and suspended in 1 mL of 40% (v/v) isopropanol. Following centrifugation (10000 g for 20 min at 4°C), pellets were washed with 2 mL of 70% (v/v) ethanol, and centrifuged at 10000 g for 20 min at 4°C. DNA pellets were left to dry for 1 hr at room temperature, and finally suspended in 50 μL of buffered (pH 8.0) deferoxamine mesylate (10 mM Tris-HCl pH 8.0, 1 mM EDTA-Na$_2$, 0.1 mM deferoxamine mesylate).

Genomic DNA (15 μg) from *P. abyssi* and *E. coli* was digested by incubation with five units of nuclease P1 (Sigma; one unit is defined as the amount of enzyme required to liberate 1.0 μmol of acid soluble nucleotides from RNA per min at 37°C, pH 5.3) for 2 hr at 37°C. Four units of alkaline phosphatase (Sigma; one unit is defined as the amount of enzyme required to hydrolyzes 1 mol of 4-nitrophenyl phosphate per min at 37°C, pH 9.8) were then added for an additional 1 hr incubation at 37°C. Released 2'-deoxyribonucleosides were centrifuged (7000 g for 5 min at 4°C) and the supernatant was recovered for injection.

## Quantification of 8-oxodG (HPLC/UV/EC)

The 8-oxodG level was determined by HPLC (Agilent 1200 series) coupled to electrochemical (Coulochem III, ESA) and UV (Agilent 1200 series) detection. The limit of detection is about 0.01 pmol of 8-oxodG (one 8-oxodG lesion per $10^6$ dG). Separation of 8-oxodG and 2'-deoxyribosides was carried out by using an Ultrasphere pre-column (5C18, Interchim) and an Uptisphere column (5ODB, Interchim). Elution was performed in isocratic mode using a mobile phase composed of 10% (v/v) methanol and 100 mM sodium acetate pH 5.2. The guard and the measure cells were respectively set to an oxidation potential of 460, 150 and 380 mV. The quantification of 8-oxodG was performed in accordance with a calibration curve previously obtained with known picomole amounts of authentic 8-oxodG. The standard expression of the number of 8-oxodG residues per $10^6$ dG, deoxyguanosine were quantified by UV detection (254 nm) of the output of the HPLC column. For the conditions described, the retention times of 8-oxodG and dG were 11.5 and 8.5 min respectively at 35°C. For each condition (stationary and exponential growth phases), the average of three measurements (n = 3) from two biological replicates was used for statistical analyses. To enumerate viable cells, most-probable-number (MPN) assays were performed as previously published (*Blodgett, 2006*).

### *P. abyssi* protein extracts

*P. abyssi* was grown in continuous culture as described in the Strains and cell culture techniques for 8-oxodG detection section of the methods. Pelleted cells were resuspended at a 1:1 (w:v) ratio in a buffer containing 350 mM MOPS pH 7.5, 6.4 mM EDTA, 5 mM DTT, 25% glycerol, 480 mM NaCl, 1 μg/ml Pepstatin A and complete EDTA-free protease inhibitor cocktail tablet (Roche). Cells were lysed by sonication using a Vibracell ultrasonic processor (BioBlock Scientific, 3 × 0.5 min followed

by 2 × 1 min, at 375 W, 40% amplitude on ice). Lysed cells were centrifuged at 10000 g for 60 min at 4°C to remove excess cell debris. The concentration of remaining protein in the cell extract was estimated using a Bradford Assay.

## Protein over-expression and purification

The proteins utilised in this study, with the exception of the p41/p46 complex, were overexpressed and purified as previously described in the following publications, PolB (*Gouge et al., 2012*), PolD (*Henneke et al., 2005*), PCNA (*Henneke et al., 2002*) and RPA (*Pluchon et al., 2013*).

For the expression of the p41/p46 complex, Rosetta 2(DE3) pLysS competent cells (Novagen) were co-transformed with the vectors pQE-80L and pET26b(+) containing p41 (Pab2236) and p46 (Pab2235) respectively. The transformed strain was grown at 37°C in LB medium supplemented with 100 μg/ml Ampicillin and 34 μg/ml Kanamycin. Protein over-expression was induced through the addition of 1 mM IPTG and left to incubate at 30°C for 4 hr before the cells were pelleted by centrifugation. Cells were resuspended in buffer A (50 mM NaP pH 6, 200 mM NaCl, 1 mM DTT, 20 mM imidazole) supplemented with complete EDTA free protease inhibitor cocktail (Roche), before being lysed by sonication. Lysed cells were treated with DNase I (Sigma-Aldrich) and incubated at 37°C for 30 min followed by heat treatment for 20 min at 75°C. Denatured *E. coli* host proteins were removed by centrifugation (20000 g for 60 min at 4°C).

Clarified lysate was loaded onto a 1 ml HisTrap HP column (GE Healthcare). Following washing, protein was eluted directly onto a 1 ml HiTrap Heparin column (GE Healthcare) using a 5 ml gradient of 0.02–1 M imidazole in 50 mM NaP pH 6, 200 mM NaCl, 1 mM DTT. Elution from the Heparin column was developed using a 5 ml gradient of 0.2–1 M NaCl in 50 mM NaP pH 6, 1 mM DTT. Primase containing fractions were pooled and loaded onto a Superdex S200 10/300 GL column (GE Healthcare) before being eluted from the column with a buffer of 50 mM MES pH 6.0, 600 mM NaCl, 1 mM DTT. Glycerol was added to a final concentration of 40% before the sample was stored at −20°C.

## Primer-template extension assays

All oligonucleotides used in this study (*Supplementary file 1*) were purchased from Eurogentec S.A. Primer-template substrates were prepared by mixing the relevant oligonucleotides at a 1:1 ratio in a buffer containing 10 mM Tris pH 8.0, 50 mM NaCl, 1 mM EDTA. Primer-template annealing was carried out by heating at 95°C for 10 min before being left to slowly cool to room temperature.

Base reaction mix was prepared with 50 nM primer-template (indicated in the relevant figure), 50 mM Tris pH 8.0, 1 mM DTT, 50 mM NaCl, 5 mM MgCl$_2$, 200 μM dNTPs. Reactions were supplemented with PCNA (100 nM) and RPA (500 nM) as indicated in the legend of each figure. The reactions mix was left to pre-incubate at 55°C for 10 min, before the reaction were initiated by the addition of the one of the polymerisation enzymes at the following final concentrations, PolB (2.5 nM), PolD (250 nM) and p41/p46 (250 nM). Reactions were left to incubate at 55°C for 5 min in the instance of single incorporation reactions, or for the time points indicated in each figure legend in the instance of time course reactions, and stopped by placing on ice and the addition of loading buffer (79% formamide, 20% glycerol, 20 mM EDTA, 1 μM reverse complement DNA).

For reactions using *P. abyssi* cell extract, 20 μg of whole cell extract was added to initiate the polymerisation reaction. The reactions were left to incubate for 60 min before being stopped by the addition of stop buffer (0.5 mg/ml Proteinase K, 5 mM EDTA, 0.5% SDS). The DNA was purified through ethanol precipitation followed by suspension in loading buffer.

In all instances, quenched reactions were denatured at 95°C for 10 min before being loaded onto an 18% denaturing acrylamide gel which was subsequently imaged using a Typhoon Scanner and analysed using ImageQuant TL 8.1 (GE healthcare) with the quantification methods as follows: Primer±n (%), densitometry measurement of primer±n as a percentage of total lane densitometry; Stall (%), densitometry measurement of identified stall band as a percentage of primer+n densitometry; 87nt (%), densitometry measurement of 87nt band as a percentage of primer+n densitometry. In all cases the background value was subtracted.

### Exonuclease degradation assays

Assays were carried out and analysed in an identical manner to that described for primer-template extension assays, but dNTPs were omitted from the reactions. Quantification of exonuclease activity (Primer-n(%)) was the densitometry measurement of all primer-n bands as a percentage of total lane densitometry following subtraction of the background value.

## Acknowledgements

This work was supported by research grant from the French National Research Agency [ANR-10-JCJC-1501–01] to Ghislaine Henneke. Tom Killelea thanks the French Institute of Marine Research and Exploitation (Ifremer) and Departmental Council of Finistère (CD29) for funding. The technical assistance of Audrey Bossé throughout this project is greatly appreciated.

## Additional information

### Funding

| Funder | Grant reference number | Author |
| --- | --- | --- |
| Agence Nationale de la Recherche | ANR-10-JCJC-1501-01 | Ghislaine Henneke |

The funders had no role in study design, data collection and interpretation, or the decision to submit the work for publication.

### Author contributions

Tom Killelea, Conceptualization, Formal analysis, Supervision, Funding acquisition, Validation, Investigation, Writing—original draft; Adeline Palud, Conceptualization, Formal analysis, Investigation, Writing—original draft; Farida Akcha, Mélanie Lemor, Conceptualization, Formal analysis, Validation, Investigation, Methodology; Stephane L'haridon, Anne Godfroy, Conceptualization, Formal analysis, Investigation, Methodology; Ghislaine Henneke, Conceptualization, Formal analysis, Supervision, Funding acquisition, Validation, Investigation, Methodology, Writing—review and editing

### Author ORCIDs

Tom Killelea https://orcid.org/0000-0003-0974-8034
Ghislaine Henneke https://orcid.org/0000-0002-2985-5526

### Decision letter and Author response

Decision letter https://doi.org/10.7554/eLife.45320.019
Author response https://doi.org/10.7554/eLife.45320.020

## Additional files

### Supplementary files

• Supplementary file 1. Conformation of primer-templates (p/t) used in this study and their oligonucleotide sequences. The location of the fluorescent labels is indicated by *.
DOI: https://doi.org/10.7554/eLife.45320.016

• Transparent reporting form
DOI: https://doi.org/10.7554/eLife.45320.017

### Data availability

All data generated and analysed in this study are included in the manuscript and supporting files.

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
