## [Decision Letter]

Thank you for submitting your article "Interplay at the replisome mitigates the impact of oxidative damage on genetic integrity of hyperthermophilic *Archaea*" for consideration by *eLife*. Your article has been reviewed by three peer reviewers, and the evaluation has been overseen by a Reviewing Editor and Gisela Storz as the Senior Editor. The reviewers have opted to remain anonymous.

The reviewers have discussed the reviews with one another and the Reviewing Editor has drafted this decision to help you prepare a revised submission. A summary below of the work is followed by major revisions and problems that the reviewers found important for final adjudication.

Killelea and colleagues investigate the ability of DNA polymerases Pol B, Pol D and the polymerase-primase p41/p46 from *Pyrococcus abyssi*, to bypass template 8-oxodG. The authors first conduct measurements to assess the levels (steady state and upon induction by oxidative stress) of 8-oxodG in the *Archaea* genomic DNA. Next the authors attempt to measure bypass in whole cell-free extracts using synthetic primer-templates. But these measurements are complicated, not surprisingly, by what appears to be repair activity in the extract. Therefore, the authors used primer extension and degradation assays with purified polymerases. The authors conclude that even in the presence of accessory proteins PCNA and RPA, Pols B and D are relatively inefficient in 8-oxodG bypass. When the polymerases pause at the lesion their exonuclease activities take over and start degrading the primer strand. The authors argue that the idling at the lesion allows displacement of the replicative pol and recruitment of p41/p46, which bypasses the lesion and in turn is replaced by the replicative polymerase.

The subject of this work is important, as so far, much less is known about TLS and repair of 8-oxodG lesions in *Archaea* relative to the knowledge of eukaryotes and bacteria. Therefore, this work will be of interest to all studying DNA replication and repair. However, some of the experiments are insufficient to support the models put forward. There are also other issues that must be addressed.

Major comments:

1) "Coupled with enhanced exonuclease activity for both Pol B and Pol D when in presence of PCNA, this raises the notion of the stalling seen when encountering 8-oxodG manifesting as prolonged idling [...]" As shown in Figure 5C, exonucleolytic degradation of the primer is stimulated by PCNA. However, these reactions are not in the presence of dNTPs, and therefore no extension or polymerase idling is possible. On the other hand, when synthesis is permitted by addition of dNTPs there is no evidence of primer degradation (Figure 3D and Figure 4—figure supplement 1B). Hence, these results do not support the authors' model.

2) There is no description of the 3´>5´ exonuclease assay in the Materials and methods section.

3) "we have identified that […] p41/p46 physically and functionally interacts with the replisome "scaffold" protein PCNA." This statement should be modified; the authors did not conduct experiments that show physical interaction between p41/p46 and PCNA.

4) "Finally, we measure the intrinsic proofreading activities" The authors measured 3´>5´ exonuclease activity and not proofreading activity, which is error correction.

5) Figure 3—figure supplement 1B, bottom of figure: "87 nt product (%)" According to Supplementary file 1, the template strand in p/t-6 substrate is a 34-mer, not a 87-mer.

Figure 3—figure supplement 1C, if this is a denaturing gel, as indicated in the Materials and methods, why does the band labeled dsDNA (34bp) migrate differently than the 34 nt marker?

Figure 2, it would be helpful if the position of the damaged base was indicated in the gels.

6) "the occurrence of TLS by PabCE can be observed with 1% of primer DNA being fully extended in oxidative damage containing substrates (Figure 2B)." What is the percentage of full-length product with the dG-containing p/t-2 substrate? This is important to assess to what extent synthesis on the p/t-2 template is hampered by the necessity to strand displace.

7) Figure 2. According to the figure legend, the survival curve is based on values from three replicate culture samples. Standard deviations should be included so that the reproducibility of the data can be evaluated. In the graph, the percentage of surviving cells appears the same or lower at T3 than T2. The authors should use a broken axis.

8) In the Results section, the statement "The cumulative effect that both accessory proteins have on primer extension is noticeable” is misleading. The results for PolB indicate that on its own, PCNA stimulates synthesis both on the damage-containing and the undamaged templates. PCNA counteracts the inhibitory effect RPA on both damaged and undamaged templates. (Figure 3D and Figure 4—figure supplement 1B).

The results of the two experiments with PolD are not consistent. In Figure 4—figure supplement 1B, the amount of primer extensions on the dG-containing template is the same with and without PCNA, but results in Figure 3D indicate that primer extension on the dG template is over 50% lower in presence of PCNA relative to reactions without PCNA. Due to this inconsistency, these results are difficult to evaluate, and the conclusions do not seem valid.

9) "Indeed, RPA not only reduces the amount of primer-n, but also the length of the final degradation product." Reduced amount of primer degradation results in longer, not shorter, degradation products. Please correct.

10) “In-vivo validation of TLS in the *Archaea* was provided through the use of primer extension by cell-free extract" Cell-free extracts are not the same as in vivo conditions.

11) A general point to be addressed perhaps in the Discussion: The manuscript starts by making the case that this organism endures the generation of a much higher level of 8-oxoG in the genome than does the mesophile *E. coli*. However, they should be cautious in the in vivo 8-oxoG measurements, which have a notorious history of over-estimation. The in vitro studies give some basic information, but more refined analysis will have to sort that out under conditions where the enzyme and substrate ratios are more realistic.

12) Introduction, second paragraph: 8-oxodG is not a base, it is a deoxyribonucleoside.

Introduction, second and third paragraphs: Remove "Ref."

13) In the experiments for which oxygen was not added and was assumed not to be present, were all materials purged with nitrogen, helium, etc. so that there was no oxygen present and were all experiments done in a glove box which was similarly purged? If so, this should be noted; if not, they should have been done that way.

14) Subsection “Genomic DNA isolation and detection of 8-oxodG”, first paragraph and elsewhere: What was the concentration of β-mercaptoethanol?

15) Subsection “Genomic DNA isolation and detection of 8-oxodG”, first paragraph: This material is not clear-enough to reproduce the procedures. It should be rewritten.

16) Introduction, third paragraph: Define the units.

17) Legend to Table 1: there are no error bars shown.

18) Figure 6: The symbols ought to be specifically defined either on the figure or in its legend. Otherwise is quite difficult to follow.

19) Supplementary file 1: The conformations are not shown and even base pairing is not shown.

20) Introduction, second paragraph: This is confusing, as MutT/MTH1 does not act on 8-oxoG in the template – maybe the authors mean 8-oxoG that is incorporated from the triphosphate and then remains in the (new) template for the next round of replication.

21) Table 1: What is the level of detection for assigning "ND" for the *E. coli* DNA samples? Meanwhile, the numbers reported for *P. abyssi* have an indefensible level of precision. I doubt that the measurements are good to one part in 10,000 (0.01%) as the numbers imply. Finally, since *P. abyssi* was grown anaerobically, it cannot be ruled out that much of the observed 8-oxoG levels are artifactual, with the lesion generated during DNA extraction, which evidently was not done under anaerobic conditions (subsection “Strains and cell culture techniques for 8-oxodG detection”).

22) Figure 2 is strange: it appears that the cells were exposed to oxygen for only the 5 minutes of treatment (but this is not clear from the figure legend or the Materials and methods section), so why, after reaching near zero at T2, do the 8-oxoG levels rise again at T3 and T4?

23) Figure 3—figure supplement 1: Probably the U residue is rapidly converted to an AP site, even (or especially) in single-stranded DNA.

24) Subsection “Bypass of 8-oxoguanine by *P. abyssi* cell extracts”: Most DNA repair enzymes act much more rapidly on lesions in double-stranded DNA than in a single-stranded context (the typical uracil glycosylase is an exception). So the explanation that the lesion is "protected" in dsDNA doesn't make sense.

25) Figure 3D/Figure 4—figure supplement 1B and associated text, and other figures and text: PCNA does not stably associate with linear DNA – it tends to slide off an end. The results would be more solid with a circular DNA substrate or one blocked at the ends; the PCNA-loading enzyme would then also be needed.

26) Figure 5: It is hard to make any conclusions about PolD, since its 3' exonuclease appears inherently very weak, despite its being present at (apparently) 100 times the PolB level and in a 5-fold excess over the amount of substrate present (assuming that the same conditions as for primer extension were used for the exonuclease assays; the Materials and methods do not seem to specify the conditions).

27) Figure 5 quantification: to determine the percentage of primer shortening shown, presumably products shorter than n-1 are included; that's the only way to account for some of the values shown. However, it is very important to correct for the extra background that comes with every band; the Materials and methods do not specify that, and some of the results look as though they might be inflated by such an artifact.

28) Discussion: In view of the foregoing concerns, some places in this section might be toned down where activities are stated to be "proficient" or "efficient".

---

## [Author Response]

Major comments:1) "Coupled with enhanced exonuclease activity for both Pol B and Pol D when in presence of PCNA, this raises the notion of the stalling seen when encountering 8-oxodG manifesting as prolonged idling […]" As shown in Figure 5C, exonucleolytic degradation of the primer is stimulated by PCNA. However, these reactions are not in the presence of dNTPs, and therefore no extension or polymerase idling is possible. On the other hand, when synthesis is permitted by addition of dNTPs there is no evidence of primer degradation (Figure 3D and Figure 4—figure supplement 1B). Hence, these results do not support the authors' model.

This section of text has been removed. As a consequence the model and its legend have been modified in order to make it consistent.

2) There is no description of the 3´>5´ exonuclease assay in the Materials and methods section.

This has now been included in the Materials and methods section.

3) "we have identified that [...] p41/p46 physically and functionally interacts with the replisome "scaffold" protein PCNA." This statement should be modified; the authors did not conduct experiments that show physical interaction between p41/p46 and PCNA.

The text has been modified to remove the word physically

4) "Finally, we measure the intrinsic proofreading activities" The authors measured 3´>5´ exonuclease activity and not proofreading activity, which is error correction.

The text has been modified and “intrinsic proofreading activity” is now replaced by “3’-5’ exonuclease activity”. When relevant, this modification has been applied throughout the manuscript.

5) Figure 3—figure supplement 1B, bottom of figure: "87 nt product (%)" According to Supplementary file 1, the template strand in p/t-6 substrate is a 34-mer, not a 87-mer.

87 nt has been corrected to 34 nt.

Figure 3—figure supplement 1C, if this is a denaturing gel, as indicated in the Materials and methods, why does the band labeled dsDNA (34bp) migrate differently than the 34 nt marker?

The gel shown is a denaturing gel. Unfortunately, fluorophores such as the positively charged Cy5 that was used in this study reduce the effectiveness of denaturing gels for denaturing fluorescently labelled dsDNA. As shown in the Materials and methods section an excess of oligonucleotide complimentary to the unlabelled template strand is added after quenching the reaction to ensure that the labelled ssDNA is unable to re-anneal. However, this strategy is less effective when using a double fluorophore substrate. To produce a decipherable image of the Cy5 labelled primer an excess of oligo complimentary to the FAM labelled template was added. To mitigate these suitable controls were added in the form of single stranded DNA size markers and a 34bp FAM labelled duplex. The figure legend of Figure 3—figure supplement 1C has been altered to explain this.

Figure 2, it would be helpful if the position of the damaged base was indicated in the gels.

The location of the damaged base has been highlighted and the legend adjusted accordingly

6) "the occurrence of TLS by PabCE can be observed with 1% of primer DNA being fully extended in oxidative damage containing substrates (Figure 2B)." What is the percentage of full-length product with the dG-containing p/t-2 substrate? This is important to assess to what extent synthesis on the p/t-2 template is hampered by the necessity to strand displace.

This information was shown in Figure 2B (Figure 2B is now Figure 3B). To facilitate a simpler comparison, it has now also been included directly in the text.

7) Figure 2. According to the figure legend, the survival curve is based on values from three replicate culture samples. Standard deviations should be included so that the reproducibility of the data can be evaluated. In the graph, the percentage of surviving cells appears the same or lower at T3 than T2. The authors should use a broken axis.

Accordingly, Figure 2 has been modified. The most-probable-number (MPN) assays were performed as previously published (Blodgett, 2006)(Oblinger, 1975). Survival (cells/ml) is based on a three-tube MNP dilution assay. Upper and lower error bars are shown. Accordingly, the text has been altered in the legend and the cited references have been provided.

For clarity, the survival curve is reported with a logarithmic y-axis which allows pointing out the differences between cell densities. No need for a broken axis.

8) In the Results section, the statement "The cumulative effect that both accessory proteins have on primer extension is noticeable” is misleading.

The statement “cumulative effect” has been deleted and the sentence is now: “The effect that both accessory proteins have on primer extension is noticeable”

The results for PolB indicate that on its own, PCNA stimulates synthesis both on the damage-containing and the undamaged templates. PCNA counteracts the inhibitory effect RPA on both damaged and undamaged templates. (Figure 3D and Figure 4—figure supplement 1B).

The results in this paragraph have been modified accordingly to this comment.

The results of the two experiments with PolD are not consistent. In Figure 4—figure supplement 1B, the amount of primer extensions on the dG-containing template is the same with and without PCNA, but results in Figure 3D indicate that primer extension on the dG template is over 50% lower in presence of PCNA relative to reactions without PCNA. Due to this inconsistency, these results are difficult to evaluate, and the conclusions do not seem valid.

The results of the two experiments with PolD are not inconsistent. Instead, they clearly matched our previous results (Henneke, 2005) showing that PolD incorporates less dNTPs in the presence of a shorter DNA primer and that PCNA stimulates full-length synthesis by PolD rather than primer utilization. (primer length of the two DNA primers used in standing and starting start reactions in this current study are exactly the same length than those in the paper of Henneke, 2005). Thus, the influence of the primer-length on primer extension again demonstrates that PolD behaves differently in the presence of primer-template with diverse length of DNA primer. This was stated in the subsection “Role of replication fork accessory proteins in replication bypass of template strand 8-oxodG“. Nevertheless, for reinforcing this peculiar behaviour, we added this comment in the text and cited our previous paper.

Accordingly to these overall comments, subsection “Role of replication fork accessory proteins in replication bypass of template strand 8-oxodG “is now modified.

9) "Indeed, RPA not only reduces the amount of primer-n, but also the length of the final degradation product." Reduced amount of primer degradation results in longer, not shorter, degradation products. Please correct.

This is amended. Indeed, reduced amount of primer degradation results in longer degradation product.

10) In-vivo validation of TLS in the Archaea was provided through the use of primer extension by cell-free extract" Cell-free extracts are not the same as in vivo conditions.

The sentence has been modified and is now:

“In this work, DNA synthesis capable of bypassing DNA lesions was measuredin vitrousing *P. abyssi* cell-extracts (*Pab*CE) from exponentially growing cells (Figure 2A). (Figure 2A is now Figure 3A).”

Besides according to this comment, we also changed the sentence at the beginning of the second paragraph of the Discussion.

11) A general point to be addressed perhaps in the Discussion: The manuscript starts by making the case that this organism endures the generation of a much higher level of 8-oxoG in the genome than does the mesophile E. coli. However, they should be cautious in the in vivo 8-oxoG measurements, which have a notorious history of over-estimation. The in vitro studies give some basic information, but more refined analysis will have to sort that out under conditions where the enzyme and substrate ratios are more realistic.

Our biological model, *P. abyssi*, has been cultivated at 95°C under anaerobic conditions. Cells have been collected under anaerobic conditions and genomic extraction has been performed according to protocols which minimise the generation of artifactual 8-oxoG such as including deferoxamine mesylate (Hofer T, et al., 2006, iol. Chem) in solutions and by avoiding the use of standard phenol/chrolorfom procedure known to provoke accidental oxidation of DNA (Claycamp HG, 1992, Carcinogenesis). This information is available in the Materials and methods section and is now better explained in the text as required in point 15.

In a very recent paper cited at the beginning of the Discussion (Barbier et al., 2016), the level of OxoG in the genome of Thermococcus gammatolerans (9.2 +/- 0.9 8-oxoG per 10exp6 dG) which thrives at 85°C under anaerobic conditions is higher than that in the genome (2.6 +/- 0.9 8-oxoG per 10exp6 dG) of the mesophile *E. coli*. Here, we also found a higher level of 8-oxoG in the genome of *Pyrococcus abyssi* compared to that in *E. coli*. Moreover, we demonstrate that the level of 8-oxoG in the genome of *P. abyssi* is higher than that in T. gammatolerans, confirming that increased temperature of growth favours DNA oxidation (95°C compared with 85°C, respectively).

Consistent with these results (our whole study) and those published previously, TLS (damage tolerance) and repair mechanisms are proposed to be active in response to DNA oxidation in the archaeal genomes.

According to these explanations and the comment of the reviewer, the first and second paragraphs of the Discussion have been modified. This is now more consistent.

12) Introduction, second paragraph: 8-oxodG is not a base, it is a deoxyribonucleoside.Introduction, second and third paragraphs: Remove "Ref."

This is amended.

13) In the experiments for which oxygen was not added and was assumed not to be present, were all materials purged with nitrogen, helium, etc. so that there was no oxygen present and were all experiments done in a glove box which was similarly purged? If so, this should be noted; if not, they should have been done that way.

*P. abyssi* cultures have been performed in a laboratory (Laboratory of microorganisms of extreme environments called “LMEE”) having more than 20 years of experience with cultivation of anaerobic hyperthermophilic strains. Continuous and batch cultivations are routinely performed in this laboratory using nitrogen-sparged gas-lift bioreactor and when required in closed vessels. Because of these anaerobic cultivation conditions, anaerobic chamber (called anaerobic glove box) equipment is a prerequisite for working with materials without oxygen. This has been applied to *P. abyssi* cultivation. So there is no doubt of preserving cells from oxygen. Nevertheless, *P. abyssi* cell cultivation, collection and storage have been described in more detail in the Materials and methods according to this comment, thus including additional relevant papers from the LMEE laboratory.

14) Subsection “Genomic DNA isolation and detection of 8-oxodG”, first paragraph and elsewhere: What was the concentration of β-mercaptoethanol?

This is now included in the corresponding Materials and methods section.

15) Subsection “Genomic DNA isolation and detection of 8-oxodG”, first paragraph: This material is not clear-enough to reproduce the procedures. It should be rewritten.

The text in this Materials and methods section has been rewritten in more detail.

16) Introduction, third paragraph: Define the units.

The enzyme unit definitions have been provided according to the enzymes’ suppliers (Sigma). These are now part of the Materials and methods section for nuclease P1 and alkaline phosphatase.

17) Legend to Table 1: there are no error bars shown.

“Error bars” has been removed. The legend of the text has been modified and only includes “standard deviation shown”.

18) Figure 6: The symbols ought to be specifically defined either on the figure or in its legend. Otherwise is quite difficult to follow.

The symbols have been defined in the figure (Figure 6 is now Figure 7).

19) Supplementary file 1: The conformations are not shown and even base pairing is not shown.

The conformation of each substrate was shown in Supplementary file 1 along with each oligonucleotide used to form it in the standard 5’ – 3’ conformation. To avoid any confusion the full sequence of each substrate is now shown with full oligonucleotides base paired.

20) Introduction, second paragraph: This is confusing, as MutT/MTH1 does not act on 8-oxoG in the template – maybe the authors mean 8-oxoG that is incorporated from the triphosphate and then remains in the (new) template for the next round of replication.

For clarity, the corresponding sentence has been removed from the text because MutT/MTH1 does not act on 8-oxoG in the DNA but rather on the precursor. Since it is not essential and has confused one reviewer, the sentence has been deleted.

21) Table 1: What is the level of detection for assigning "ND" for the E. coli DNA samples?

The limit of detection of the HPLC/EC/UV is 0.01 pmol of 8-oxodG which corresponds to one 8-oxodG lesion per 10^6^ dG. To facilitate a simpler comparison between *E. coli* and *P. abyssi*, it has now also been included directly in the text of the Materials and methods and in the legend of Table 1.

Meanwhile, the numbers reported for P. abyssi have an indefensible level of precision. I doubt that the measurements are good to one part in 10,000 (0.01%) as the numbers imply.

All data are reported as mean ± standard deviation (SD), with *n*=3. *n*=3 refers to three independent DNA extraction from either exponentially or stationary phase cells. In addition, two different biological samples for each phase cells have been measured. Bases on this set of measurements, the mean and standard deviation have the same unit (0.01) of precision which does not seem to be unusual according to the literature. Indeed, there are several publications in which the levels of 8-oxo-dG lesion per 10^6^ dG reached the same level of precision (Speina, E. et al., 2005, Journal of the National Cancer Institute; Roszkowski, K., 2012, Cancer Epidemiol Biomarkers Prev.; Guindon-Kezis, K.A., 2014, Toxicology).

Nevertheless, to avoid any astonishment and confusion by the readers, the precision of reported data now corresponds to one digit after the decimal point (0.1).

Finally, since P. abyssi was grown anaerobically, it cannot be ruled out that much of the observed 8-oxoG levels are artifactual, with the lesion generated during DNA extraction, which evidently was not done under anaerobic conditions (subsection “Strains and cell culture techniques for 8-oxodG detection”).

This comment overlaps point 11. As explained, genomic extraction and preparation have been performed according to protocols which minimise the generation of artifactual 8-oxoG such as including deferoxamine mesylate (Hofer T, et al. 2006, iol. Chem) in solutions and by avoiding the use of standard phenol/chrolorfom procedure known to provoke accidental oxidation of DNA (Claycamp HG, 1992, Carcinogenesis). Besides, our methodology is consistent with a recent paper (Chepelev, NL., 2015 J. Vis. Exp) describing a detailed protocol for the detection of 8-oxo-dG by HPLC-EC-UV in DNA from cell cultures. This methods paper illustrates how DNA sample preparation should be achieved to minimize undesirable DNA oxidation that can occur during sample preparation (Chepelev, NL., 2015 J. Vis. Exp).

On the other hand, the values of 8-oxodG in the genome of *P. abyssi* are clearly consistent with those determined recently in the genome of a closely related specie *T. gammatolerans* (Barbier et al., 2016). Interestingly, upon exposure to oxidative stress a two-fold increase of 8-oxodG is measured in the genome of *T. gammatolerans* and *P. abyssi*, respectively.

Thus, our employed method successfully analyses DNA samples by minimizing the introduction of artifactual DNA oxidation. It provides reliable detection and quantification of 8-oxodG in the genome of *P. abyssi*.

22) Figure 2 is strange: it appears that the cells were exposed to oxygen for only the 5 minutes of treatment (but this is not clear from the figure legend or the Materials and methods section), so why, after reaching near zero at T2, do the 8-oxoG levels rise again at T3 and T4?

The time of exposure to oxygen followed by recovery of anaerobic cultivation is now clearly reported in the graph and the text. Time of oxygen exposure was 5 minutes followed by recovery of anaerobic cultivation by nitrogen sparging to the cultivation. Nevertheless, the complete anaerobic recovery is not immediate and requires 20 minutes upon nitrogen sparging. It is now better explained and illustrated in the legend of Figure 2 but also in the corresponding Results section entitled (Rate of 8-oxodG in the genome of *P. abyssi*).

At T3 and T4, the level of 8-oxodG has been determined from the genome of cells which survived the oxidative stress. This demonstrates the recovery of cells by increased cell densities along with the recovery of the basal level of 8-oxodG in their genome (respectively, 65.1 and 77.11 8-oxoxdG per 10exp6dG at T3 and T4). These values are consistent with those found at T0 and in Table 1 (60.1+/-15.9 and 63.2+/-4.6 of 8-oxoxdG per 10exp6dG, respectively). According to the mean+/-standard deviation found at T0 and Table 1, the values obtained at T3 and T4 are considered similar and consistent with the basal level.

At T2, the level of 8-oxodG was not measurable because of very low cell densities, thus preventing a sufficient amount of extracted genomic DNA for the detection by HPLC-EC-UV. It is now better explained in the corresponding Results section entitled (Rate of 8-oxodG in the genome of *P. abyssi*).

23) Figure 3—figure supplement 1: Probably the U residue is rapidly converted to an AP site, even (or especially) in single-stranded DNA.

This may well have occurred, but it is impossible to verify from the available data. Moreover, part C of the figure clearly shows that cleavage of both dU and 8-oxodG containing templates producing FAM labelled products <10 nt in length, thus highlighting phosphodiester cleavage. As such this is the focus of the text.

24) Subsection “Bypass of 8-oxoguanine by P. abyssi cell extracts”: Most DNA repair enzymes act much more rapidly on lesions in double-stranded DNA than in a single-stranded context (the typical uracil glycosylase is an exception). So the explanation that the lesion is "protected" in dsDNA doesn't make sense.

The word “protected” has been removed and the text adjusted accordingly

25) Figure 3D/Figure 4—figure supplement 1B and associated text, and other figures and text: PCNA does not stably associate with linear DNA – it tends to slide off an end. The results would be more solid with a circular DNA substrate or one blocked at the ends; the PCNA-loading enzyme would then also be needed.

The use of linear DNA substrates to study the impact of PCNA:DNA polymerase interactions has previously been used in a number of peer reviewed publications by not only our research group, but also others (Crespan,E., ACS Chem Biol., 2013; Yan, J., Nature Comm, 2017; Narita, T.Genes to cell, 2010). For instances where data shows the impact of PCNA on 8-oxodG bypass by the DNA polymerases a control containing just dG in the place of 8-oxodG is also included so that any “sliding off” can be taken into account. Moreover, the data present here is fairly conclusive in regard to PCNA impacting on the DNA polymerase performance, suggesting that “falling off” is less problematic than implied in this comment.

26) Figure 5: It is hard to make any conclusions about PolD, since its 3' exonuclease appears inherently very weak, despite its being present at (apparently) 100 times the PolB level and in a 5-fold excess over the amount of substrate present (assuming that the same conditions as for primer extension were used for the exonuclease assays; the Materials and methods do not seem to specify the conditions).

The Materials and methods have been modified to include the exonuclease assay conditions as also asked in Point 2. Enzyme concentrations used to produce this data were chosen after assaying polymerisation over 30 minutes, with concentrations of Pol B, PolD and P41/P46 used that exhibit comparable levels of polymerase activity when extending dG containing p/t-1. This strategy was chosen to avoid any discrepancies that could arise in protein purifications and to ensure that the results were not biased by the inherent differences in polymerase activity of what are three very different DNA polymerising enzymes. While it is true that the data highlights that PolD is for the most part a weaker exonuclease than PolB it can clearly be seen from Figure 5 (now Figure 6) that the exonuclease activity of PolD on a dT:dG mismatch is effective, suggesting that the substrate is important to its specific activity. As such we don’t believe that it is accurate to describe the activity as inherently weak.

27) Figure 5 quantification: to determine the percentage of primer shortening shown, presumably products shorter than n-1 are included; that's the only way to account for some of the values shown. However, it is very important to correct for the extra background that comes with every band; the Materials and methods do not specify that, and some of the results look as though they might be inflated by such an artifact.

The Materials and methods have been modified to include exonuclease quantification.

When quantifying the data all care was taken to correct for background interference. Considering the reviewers concern about this particular figure the quantification has been repeated with the new quantification values included in the figure (Figure 5 is now Figure 6). The re-quantification shows no significant variation (+/- 5%) from that originally shown with the exception of result for PolD +PCNA acting on p/t-4 (dT:dG) Figure 5C, which has been altered from 62% primer-n to 45%.

28) Discussion: In view of the foregoing concerns, some places in this section might be toned down where activities are stated to be "proficient" or "efficient".

Accordingly, the text has been modified in the third paragraph (“proficient” deleted in the fourth sentence) and in the fourth paragraph of the Discussion (“proficient” and “efficient” have been toned down).